



# Strateole 2 balloons reveal persistent errors in reanalyzed winds and trajectory calculations in the tropical lower stratosphere

Pierre Cadiou[1], Riwal Plougonven[2,3], Aurélien Podglajen[2,4,6], Albert Hertzog[2,5], and Alexandra Mac Farlane[2,3]

[1]Météo France
[2]Laboratoire de Météorologie Dynamique / IPSL, Paris, France
[3]Ecole Polytechnique, Institut Polytechnique de Paris, Paris, France
[4]CNRS, France
[5]Sorbonne University, Paris, France
[6]Ecole Normale Supérieure, PSL University, Paris, France

**Correspondence:** Riwal Plougonven (riwal.plougonven@lmd.polytechnique.fr)

**Abstract.** Winds in the tropical lower stratosphere raise difficulties for numerical weather prediction models: without geostrophy, winds decouple from temperature and direct observations are scarce. The Strateole 2 project explores the tropical lower stratosphere using superpressure balloons that drift for up to three months between 18 and 21 km altitude. Wind is measured on all flights: eight in the first campaign (2019-2020) and seventeen in the second (2021-2022). These measurements are used

to assess errors in the winds of the ERA5 reanalysis for latitudes between 18°S and 10°N. Two additional objectives of this study are to assess errors in modelled balloon trajectories, and to document the dispersion of air below the balloons, in order to facilitate the interpretation of observations made below the balloons. The comparison of measured and reanalyzed winds reveal significant errors, with standard deviations of $3.76\,\mathrm{m\,s^{-1}}$ for zonal and $3.24\,\mathrm{m\,s^{-1}}$ for meridional wind. Relative to a previous comparison in 2010, only a modest decrease of 20 and 10% is found. Trajectory calculations have very variable skill, with

median errors after 24 hours of 260 km, but a tenth of the errors larger than 600 km. Factors leading to large errors, such as initial wind error and latitude are identified. Similarly, trajectory dispersion of air below the balloon is very variable, depending on the initial shear. The persistent errors highlight the need for regular obsevations of winds in the tropical lower stratosphere, and emphasize the need for caution when using trajectory calculations for process studies.

## 1 Introduction

The tropical Upper Troposphere and Lower Stratosphere (UTLS) is a key region for the atmospheric circulation and climate. The overturning circulation of the stratosphere, i.e. the Brewer-Dobson circulation (Butchart, 2014), involves a slow entry of air from the troposphere at tropical latitudes, different paths from the tropics to higher latitudes and descent back to the troposphere. The characteristics of air entering the stratosphere, in particular its water vapor content and concentrations of trace constituents, is therefore set in the tropical tropopause region, which acts as a *gateway*. In contrast to the extra-tropics,

the transition from tropospheric characteristics of the air to stratospheric characteristics gradually occurs over a layer of several kilometers, from about 150 hPa (or 14 km) to 70 hPa (or 18.5 km), in the *Tropical Tropopause Layer* (Fueglistaler et al., 2009).



The dynamics of this region and the processes at play in determining the atmospheric composition are complex and difficult to model (Randel and Jensen, 2013), because of the multiplicity of the phenomena involved and their vast range of scales: from equatorial waves having planetary horizontal extent but vertical wavelengths of a few kilometers or less (e.g. Bramberger et al., 2023), to convective overshoots locally inserting very different air (e.g. Behera et al., 2022). Yet, the composition of the tropical lower stratosphere is of importance for climate and the radiative budget (Riese et al., 2012). Among constituents, water vapour has a major importance because of its importance for the radiative budget (Solomon et al., 2010). In situ cirrus clouds constitute a key step in the dehydration of air parcels entering the stratosphere (Fueglistaler et al., 2009). Lagrangian trajectory calculations are an important tool in quantifying the combination of temperature fluctuations and advection leading to their formation (e.g. Gettelman et al., 2000; Jensen and Pfister, 2004; Jensen et al., 2012; Schoeberl et al., 2016). These process studies however crucially rely on reanalyzed winds for the quasi-horizontal part of their advection (e.g. Inai et al., 2013).

The equatorial upper-troposphere and lower-stratosphere also remain challenging regions for numerical weather prediction models (Baker et al., 2014; Friedrich et al., 2017). Improved modeling of the stratosphere is beneficial for forecasts on sub-seasonal to seasonal timescales (Lee et al., 2018; Scaife et al., 2022), as the stratosphere is a source of predictability on these timescales (Vitart and Robertson, 2018). Measurements of winds in the tropical upper-troposphere and lower-stratosphere are particularly scarce (Baker et al., 2014), and additional observations are of significant value (Rennie et al., 2021). The ADM-Aeolus Doppler satellite lidar instrument has flown from 2018 to 2023 (McMahon, 2019), providing rare and precious measurements on the global winds, notably in the tropics. At lower stratospheric heights, long-duration superpressure measurements provide very valuable in situ measurements, with an extensive spatial coverage but only for the months of the campaigns. The technological campaign of 2010 has already provided a rare opportunity to assess the accuracy of analyses and reanalyses (Podglajen et al., 2014a). This highlighted significant errors, with standard deviations more than $3\,\mathrm{m\,s^{-1}}$. A new opportunity is provided by more recent balloon campaigns, for which the opreational phase has begun in 2019.

The Stratreole 2 project consists of three campaigns aiming to observe and better understand processes in the equatorial upper troposphere and lower stratosphere, from dynamics and transport to cirrus clouds and aerosols (Haase et al., 2018). The long-duration superpressure balloons used drift with the winds for up to three months each. Filled with helium, their inextensible envelope ensures a constant volume once flight level is reached, and the balloons thus behave essentially as ispopycnic tracers. Two flight levels are used: balloons drifting near 68 hPa are referred to as *TTL* balloons (Tropical Tropopause Layer), and those drifting near 50 hPa are referred to as *STR* balloons (Stratosphere). Eight balloons flew during the technological campaign, from October 2019 to February 2020, and 17 balloons flew during the first scientific campaign, from October 2021 to January 2022. The second scientific campaign is currently planned for the boreal fall-winter period of 2025. These stratospheric balloon campaigns extend previous investigations at high latitudes (Hertzog et al., 2007; Rabier and coauthors, 2010).

The first objective of the present study is to use the Strateole 2 measurements to assess the accuracy of winds in the ERA5 reanalyses. We choose to focus on winds because the errors and biases are more important than for temperature (Podglajen et al., 2014a; Kawatani et al., 2016), and we focus on the ERA5 reanalysis as it is widely used and constitutes one of the more accurate reanalyses (Fujiwara et al., 2022). In complement to documenting errors in the reanalyzed winds, errors in balloon trajectories calculated from reanalyzed winds are also investigated (Selvaraj et al., 2019).



The second objective of the present study is to investigate and document air motions relative to the balloon, in order to ease the interpretation of certain measurements. Indeed, many different instruments have flown on the different configurations of Strateole 2 balloons, targeting in particular water vapor, trace constituents and aerosols at the balloon level and below. The instruments notably include a downward looking lidar (Lesigne et al., 2023) and a reeled-down instrument (Kalnajs et al., 2021). Investigation of air motions around the balloon are needed to better interpret these measurements and have insights of the origin of the air sampled. The balloon-borne measurements indeed sit between two extremes: when a cirrus cloud is observed by satellite (e.g. Winker et al., 2009) or airborne instruments (e.g. Jensen et al., 2017) the measurement provides a vertical curtain for one snapshot in time. In contrast, theoretical studies using box models follow hypothetical air parcels in a purely Lagrangian way (e.g. Jensen and Pfister, 2004; Spichtinger and Krämer, 2013; Schoeberl et al., 2016). The observations of cirrus clouds by a balloon-borne lidar provide a spatio-temporal sampling that is intermediate: in consequence, the interpretation in terms of time evolution of the cirrus, or of spatial variations of it, requires a quantification of the relative motions of the air at the balloon level and below.

The paper is organized as follows: the observations and the trajectory calculations are presented in Sect. 2. The wind measurements and real balloon trajectories are used to assess the accuracy of reanalyzed winds and of calculated trajectories in Sect. 3. The motion of air parcels relative to the balloons are investigated in Sect. 4. The results are summarized in Sect. 5 and conclusions given in Sect. 6.

## 2 Data and methodology

### 2.1 Balloon flights and observations

The respectively 8 and 17 superpressure balloon trajectories of the 2019 and 2021 Strateole-2 campaigns are displayed in Fig. 1. In both cases, the balloons were launched from Mahé international airport, Seychelles (55.5º E, 4.7º S), and the flights mostly occurred in the deep tropics. Note that the CNES (*Centre National dÉtudes Spatiales*, the French space agency) superpressure balloons used during Strateole-2 are freely advected by the winds once at float: unlike Loon balloons (e.g., Schoeberl et al., 2017), no manoeuvering device able to change the balloon altitude is carried by Strateole-2 balloons. Superpressure balloon flights typically last for several weeks, which enable the balloons to travel long distances. The flight durations were generally longer during the 2019 campaign, and some of the 2019 ballons actually achieved a full circumnavigation of the Earth. The longest flight lasted for 107 days.

11-m and 13-m diameter balloons were used during Strateole-2. They are associated with flight altitudes 1.5 km apart: typically $18.5 - 19$ km (70 hPa) and $20 - 20.5$ km (50 hPa) respectively. Balloons flown at the lower altitude are referred to as 'TTL' (Tropical Tropopause Layer) flights, while those flown at the upper altitude are referred to as 'STR' (Stratosphere) flights. Both campaigns took place during a westerly phase of the Quasi-Biennial Oscillation (QBO).

Each of the Stratéole-2 balloons carries a GPS receiver and the TSEN instrument, which provides in-situ observations of air pressure and temperature. The balloon position and the air temperature are measured every 30 s along the trajectory, while the air pressure is measured every 1 s. The balloon horizontal positions and air pressure are respectively measured with a typical





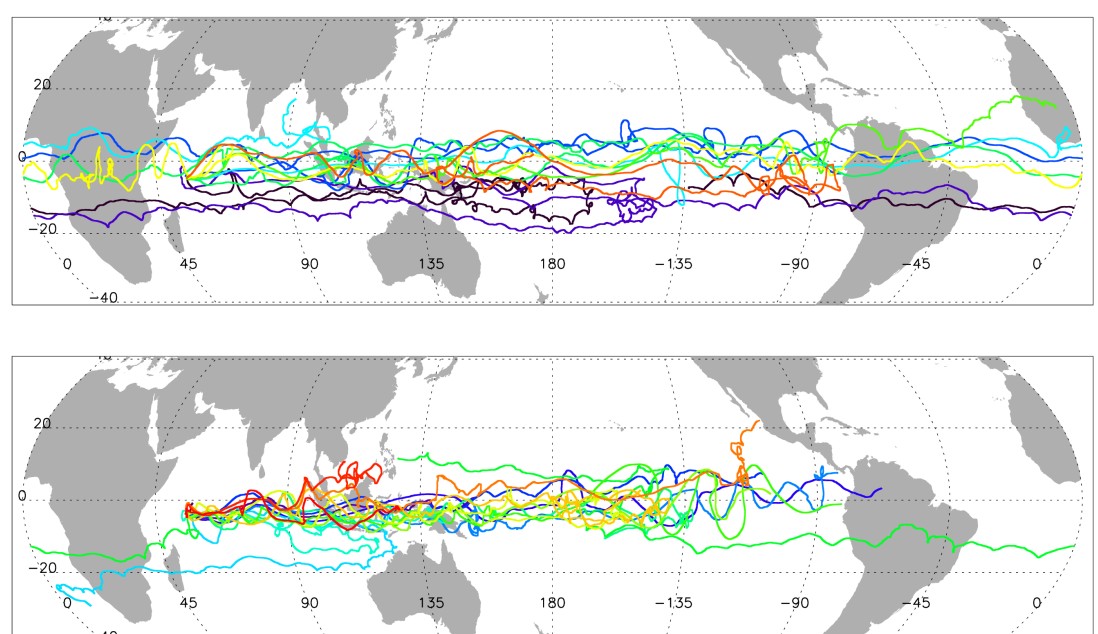

**Figure 1.** Trajectories of Strateole-2 balloon flights during (top) the 2019 and (bottom) the 2021 campaigns.

accuracy of 1.5 m and 10 Pa. Centered finite differences are used to compute the balloon winds from the balloon positions
every 30 s. Since atmospheric drag is the dominant horizontal force acting on the balloon, balloon winds provide an excellent
estimate of the air-parcel zonal and meridional velocities: typical differences are less than $0.1\,\mathrm{m\,s^{-1}}$ (Podglajen et al., 2014a).
In the vertical direction on the other hand, buoyancy is the dominant force, and superpressure balloons oscillate about their
equilibrium density surface (Vincent and Hertzog, 2014). At first order then, superpressure balloon trajectories are isopycnic.

**2.2 Trajectory calculations**

Balloon-borne winds and trajectories are primarily compared to ECMWF ERA-5 reanalyses. Comparisons with ECMWF IFS
forecasts will also be briefly presented. A first assessment of model errors is performed by directly interpolating the model
winds into the balloon positions. For this, we use a cubic-spline interpolation in longitude, latitude, time and log-pressure of
the tri-hourly model winds, which were retrieved on a $0.125^{\mathrm{o}} \times 0.125^{\mathrm{o}}$ grid and on the 137 ECMWF model levels. The time
resolution of the modelled winds was tested, using hourly resolution for calculation of the trajectories in 2019, and found to
have little impact.

The trajectory comparison is achieved by computing offline trajectories in model winds. The trajectory code is the same as
that described in Podglajen et al. (2020). Namely, it uses a $4^{\mathrm{th}}$-order Runge-Kutta scheme with an adjustable time step to advect
the trajectories in model fields. The interpolation module is the same as for the wind comparison. The trajectory results are
ouput every 10 min. The trajectory code can be integrated either forward or backward in time, and can use ERA-5 reanalyses or



IFS forecasts as inputs. In what follows, we focus on trajectories calculated from the reanalyses, and refer to those as *modelled* trajectories or simply trajectories. Those calculated from forecasts will be referred to as forecast trajectories. The latter are notably used during balloon operations, to anticipate instrument operations and flight management.

Given a set of modelled wind, pressure and temperature fields, two types of trajectories were computed. First, constant-density (or isopycnic) trajectories mimic the real balloon behavior. The densities used for those trajectories are therefore those of real balloon flights. Computations were started from each balloon's position every three-hours following a isopycnic trajectory for eight days. For the 2019 campaign, we additionally launched isentropic trajectories, with the same initial positions, to investigate the divergence of the balloon motions with respect to those of air parcels. Second, we have also computed isentropic trajectories, which are closer to air-parcel motions in the atmosphere, launched a number of altitudes below

and above the balloons. The goal is to study the movement of the air measured by some instruments below the balloon. We launched such trajectories every 12 hours on 20 points from 48 hPa to 100 hPa at the longitude and latitude of the balloon, for 3 day isentropic back trajectories. These provide some information on the origin of air parcels sampled by profiling instruments carried during Strateole-2.

## 2.3 Errors and diagnostics

The evaluation of trajectories involves calculation of errors in position after different time intervals: one, two or three days. These are calculated as distances on the Earth between the real and the modelled positions for the chosen balloon, using the Python package *geopy* (Gonzalez-Nieto et al., 2020). In addition to trajectories calculated at the level of the balloon, isentropic trajectories of air below the balloons are also calculated. The purpose is to document the origin of air in which phenomena are observed by certain Strateole 2 instruments. For these trajectory calculations, there are no error calculations as we do not have

observations of real air trajectories. We quantify the dispersion of air relative to a reference level - this can be thought of as the level from which observations are carried out. These calculations introduce several parameters and the following choices have been made:

－ as a metric for the vertical separation of back trajectories, we choose to use potential temperature as the main coordinate for vertical level, as it is a conserved quantity along adiabatic motions, and pressure as a complementary coordinate as it

is very commonly used.

－ To quantify how much back trajectories separate spatially, we introduce

$$D_{\theta_r}(t, \theta) = \text{distance after time } t \text{ between isentropic trajectories at levels } \theta \text{ and } \theta_r, \tag{1}$$

where $\theta_r$ is a chosen reference level.

－ for practical purposes, the rate of separation of back trajectories in time are somewhat difficult to interpret as growth

rates (equivalent to a velocity). It is found preferable to converte them to a distance, i.e. the distance covered in one day at that velocity:

$$\tilde{d}(t, \theta) = \frac{D_{\theta_r}(t, \theta)}{t} 1d. \tag{2}$$





In other words, $\tilde{d}$ describes the expected distance after one day for a separation occuring at the average velocity difference between the two levels.

The assessment of the dispersion of back trajectories will reveal a large variability from one situation to another. Identifying factors that are indicative of this dispersion will prove useful: the vertical shear at the reference time will prove meaningful. Hence the separation distance between two back trajectories is rescaled with the distance that can be expected from shear between the two levels and for the reference time $t_0$ of these back trajectories (date from which the back trajectories are *'started'*):

$$\hat{D} = \frac{D_{\theta_r}(t, \theta)}{\Lambda |t_0 - t|},\tag{3}$$

where $\Lambda$ is the shear present at the reference time of the backtrajectory between levels $\theta_r$ and $\theta$.

## 3  Assessment of the realism of reanalyzed winds

The present section investigates the errors in reanalyzed winds, both in instantaneous comparisons (Sect. 3.1) and through their integrated effect over time in calculated trajectories (Sect. 3.2). The errors in forecast trajectories are also briefly discussed

(Sect. 3.3).

### 3.1  Errors in reanalyzed winds

As a preliminary to the systematic characterization of the errors, two illustrative portions of trajectories are displayed in Fig. 2. Modelled trajectories, calculated using the reanalyzed wind fields, can provide a striking, visual illustration of the variable quality of reanalyzed winds (Selvaraj et al., 2019). The upper panel illustrates an example of a modelled trajectory which

closely follows the real balloon trajectory. Differences are present in the wind fluctuations, but the relevance of the trajectory calculation over eight days clearly suggests that errors in reanalyzed winds, in this example, are weak. The second panel of Fig. 2 shows a contrasting example of diverging trajectories. To illustrate that the divergence of the modelled and real trajectories is not an isolated behavior, the first 24 hours of subsequent trajectories (every 24 hours) are also plotted, confirming that the errors in the modelled winds persist for several days.

These two trajectories illustrate the range of situations encountered regarding lower stratospheric winds in reanalyses: there is a stark contrast between situations for which errors are weak, as is common and expected in the extra-tropics and high latitudes, with errors being unbiased and weak (Selvaraj et al., 2019), and situations where the errors can be of the same order as the winds. The errors on the winds are quantified systematically below.

Errors in the zonal and meridional winds of the ERA-5 reanalysis were quantified by interpolating every three hours the

reanalayzed winds to the positions of the balloons. Results were analyzed separately for the measurements of both campaigns, 2019 and 2021, but turned out to be very similar. Hence only the errors for the whole data set are presented. The Probability Density Functions (PDF) of the errors on zonal and meridional wind are displayed in Fig. 3, with key characteristics listed in table 1 and 2. Salient features are as follows:





(a) $25^{th}$ of January 2020

(b) $8^{th}$ of December 2019

**Figure 2.** Eight day trajectory and modelled trajectory of balloon 1 starting the $25^{th}$ of January 2020 (a) and the $8^{th}$ of December 2019 (b). There is a dot every 24 hours on the trajectory.

1. reanalyzed zonal winds display a significant negative bias of $-2.08$ m.s$^{-1}$, whereas meridional winds are unbiased. This bias was already highlighted from the wind observations of Pre-Concordiasi in 2010 (Podglajen et al., 2014a). It is indicative of westerly winds that are underestimated by the model. The bias on ECMWF Operational Analysis' zonal wind was $-2.4$ m.s$^{-1}$ during Concardiasi. Note that the three superpressure balloon campaigns sampled essentially the same phase of the QBO, ie the westerly phase and the beginning of the transition to the easterly phase.

2. The standard deviation of meridional wind for the Strateole campaigns is 3.24 m.s$^{-1}$. For comparison, it was 3.8 m.s$^{-1}$ for the ERA-Interim reanalysis (Dee et al., 2011) meridional wind during the Pre-Concordiasi campaign (Podglajen et al., 2014a). This suggests a significant decrease, although errors remain large.





3. The standard deviation of zonal wind is 3.76 m.s$^{-1}$, whereas it was 4.8 m.s$^{-1}$ for ERA-I during Pre-Concordiasi (Podglajen et al., 2014a). This constitutes, as far as the data may be representative, a significant improvement of more than 20%.

4. Both distributions appear visually comparable to a normal distribution: a Gaussian distribution with the same mean and standard deviation is plotted on both panels for comparison. However, several statistical tests (Lilliefors, Shapiro-Wilk, Anderson-Darling (Mohd Razali and Yap, 2011)) establish that neither zonal nor meridional wind error exactly follow a normal distribution. Indeed, both wind distributions have a positive excess kurtosis (0.2 for zonal wind error and 0.4 for meridional wind error) which means that the distribution tails are heavier than for a normal law. Moreover, the distribution of errors for zonal wind has negative skewness, indicating an asymmetry with more errors on the right of the mean. These characteristics, will be more noticeable in the next section.

The most striking difference between the errors in zonal and meridional winds is the bias in the zonal wind, which has no counterpart for meridional wind and calls for careful consideration. Indeed, part of the bias in zonal wind arises as an interesting consequence of quasi-Lagrangian sampling. First, consider the expected signature of different families of waves with small amplitude, say on the meridional wind: they induce alternately positive and negative meridional wind values, which are symmetrically sampled by the balloon. Now, among equatorial waves, Kelvin waves have specificities: they have no signature on the meridional velocity, and they only propagate eastward. Additionally, they attain rather large amplitudes. The anomalies in zonal wind of Kelvin waves with finite amplitudes will be sampled asymmetrically: when a balloon is in the positive $u$ anomaly, the balloon moves in the same direction as the phase speed of the wave. The balloon stays longer in the positive anomaly. Reversely, the negative $u$ anomaly shortens the stay of the balloon in the opposite phase of the Kelvin wave. For large-amplitude waves, the balloon may *surf* in the positive phase of the wave (Podglajen et al., 2014a). Hence, the significant bias in zonal wind is in part a signature of missing Kelvin wave activity in the reanalysis. It is known that Kelvin waves in analyses or reanalyses are underestimated (Kim and Alexander, 2015). In Appendix A virtual balloon trajectories are used to estimate a lower bound for the contribution of this effect. It is found to be slightly larger than $1\,\mathrm{m\,s^{-1}}$, suggesting that this may account for a major part of the bias found in zonal wind.

Regarding the variability of the winds, the diagnostics are less contrasted than for the bias: the standard deviation for errors on zonal wind are larger than those for meridional winds (tables 1 and 2). In contrast, the correlation coefficient between modelled and observed winds is 0.87 for zonal winds and only 0.72 for meridional winds. This difference is a priori mainly due to the different phenomenology for zonal and meridional winds, the latter being dominated by wave perturbations mainly on shorter time and space scales than zonal winds.

This different phenomenology between zonal and meridional winds is further illustrated by the relationship between the amplitude of the wind components and the corresponding errors, shown in Fig. 5. For the meridional wind, panel *b)* shows a strong correlation between the wind amplitude and the error, with the errors approximately half of the meridional wind (slope of $-0.48$). This is consistent with the interpretation that the analyses describe a significant part of the equatorial waves present but underestimate the amplitudes (Kim and Alexander, 2015). For the zonal wind (panel *b)*), this relation is also present but



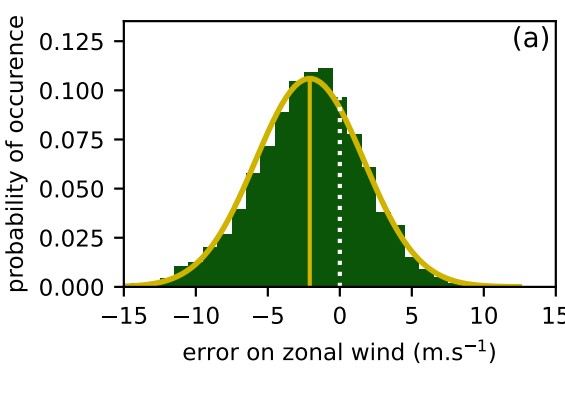

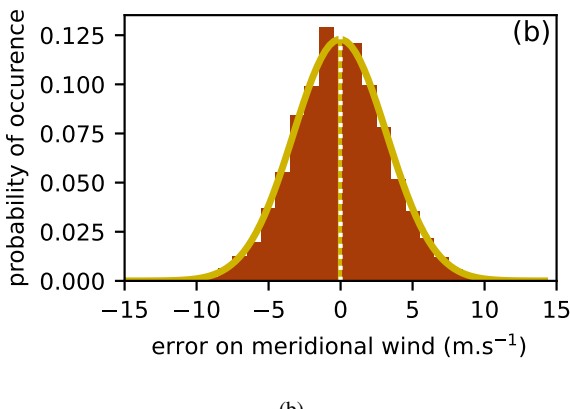

(a)                                                                 (b)

**Figure 3.** Histograms of ERA-5 wind error from three-hourly data of every balloon from both 2019 and 2021 Strateole campaign. Solid orange line is the Gaussian curve with the same mean and standard deviation as the histogram. Dashed yellow line is the mean. Dotted white line indicates the 0 of the horizontal axis. Bin size for both panels is $1\,\mathrm{m\,s^{-1}}$

.

| | Pre-concordiasi | 2019 Strateole campaign | 2021 Strateole campaign | 2019 and 2021 campaign |
|---|---|---|---|---|
| mean | −2.4 | −2.15 | −2.03 | −2.09 |
| standard deviation | 4.8 | 3.89 | 3.62 | 3.76 |

**Table 1.** Mean and standard deviation of zonal wind error across different campaigns

less strong as there are errors also tied to the mean winds and to the Kelvin waves. The slope of the regression is only 0.25 and there is significantly more scatter.

Regarding the geographical distribution of errors, wind measurements from the 2010 campaign had suggested a strong
longitudinal variation of errors, with strongest errors over the Indian Ocean and East Pacific (see Podglajen et al. (2014a), and in particular Fig. 10). Although the sampling by only three balloons was limited, this evidence was consistent with the paucity (or absence) of observations in these regions and with other lines of evidence (Baker et al., 2014). Now, the more numerous

| | Pre-concordiasi | 2019 Strateole campaign | 2021 Strateole campaign | 2019 and 2021 campaign |
|---|---|---|---|---|
| mean | 0.00 | −0.00 | −0.08 | −0.04 |
| standard deviation | 3.6 | 3.44 | 3.04 | 3.24 |

**Table 2.** Mean and standard deviation of meridional wind error across different campaigns



observations from the first two Strateole 2 campaigns do not reproduce this conspicuous longitudinal variation of errors in the winds, as shown by Fig. 4. Although the RMS error of zonal wind is greater over the Indian and Eastern Pacific ocean than over the maritime continent the contrast is not as large as it was in Podglajen et al. (2014a). It also appears there is a lack of balloon's observations between 80°W and 40°E which limits the study of data over South America and Atlantic Ocean. The longitude does not seem to have any significant influence on meridional wind error. Several factors likely contribute to these differences:

- the sampling in 2010 was very limited (three balloons, and in fact mainly two of those), and within this sparse sampling several striking episodes of strong errors occurred (of course in the Indian Ocean and Eastern Pacific), as documented in Podglajen et al. (2014a);

- the model used for the ERA5 reanalysis has changed and improved relative to the version used in operational analyses in 2010. Some of these changes may have contributed to a better representation of the dynamics in the equatorial belt, and a more physical relation between the tropospheric forcing and resulting equatorial waves in the upper troposphere and lower stratosphere. The representation of these waves has certainly improved, be it only with the enhanced resolution. Such improvements could in principle contribute to reduce the dependency on observations for the lower-stratospheric winds, and hence reduce the contrast between different longitudes. However, such hypothesis is not quite compatible with the modest improvement of the overall errors in winds.

One should also keep in mind that the Aeolus wind measurements, which would improve lower stratospheric winds throughout the tropical belt (Bley et al., 2022; Pourret et al., 2022; Laroche and St-James, 2022), are not assimilated in ERA5.

## 3.2 Errors in modelled trajectories

The errors in modelled trajectories are closely related to the errors in the wind, but they are of interest for at least two reasons: for practical purposes they are very close to trajectory forecasts which are an indispensable operational component of balloon campaigns for safety. More fundamentally, errors in trajectories inform us on the coherency over time of the errors that were identified in the instantaneous comparison of analyzed to observed winds.

Figure 6 displays the distribution of errors in the modelled trajectories after either 24 or 48 hours: in longitude, in latitude and as a distance to the real position. A first result is that the distributions resemble the distributions for the error on the wind: the error in longitude includes a negative bias, and has a wider distribution than the error in latitude, which is unbiased.

Note that we express errors in longitude and latitude in degrees: one may simply retain an approximate value of 110 km to convert an error in latitude or longitude (in degrees) to an error in kilometers. Indeed, as the majority of observations was collected within 10 degrees of the Equator, the effects of spherical geometry remain negligible: a degree in longitude decreases from about 111.1 km at the Equator to about 109.4 km at 10° of latitude.

The bias in longitude can be compared to the bias in instantaneous values of the wind. The bias in the zonal wind was found to be $-2.09\ \mathrm{m\,s^{-1}}$. If errors on the zonal wind persist coherently for times of a day or longer, one expects the distribution of errors in longitude to be approximately close to a scaled version of the distribution of errors in zonal wind. The bias would





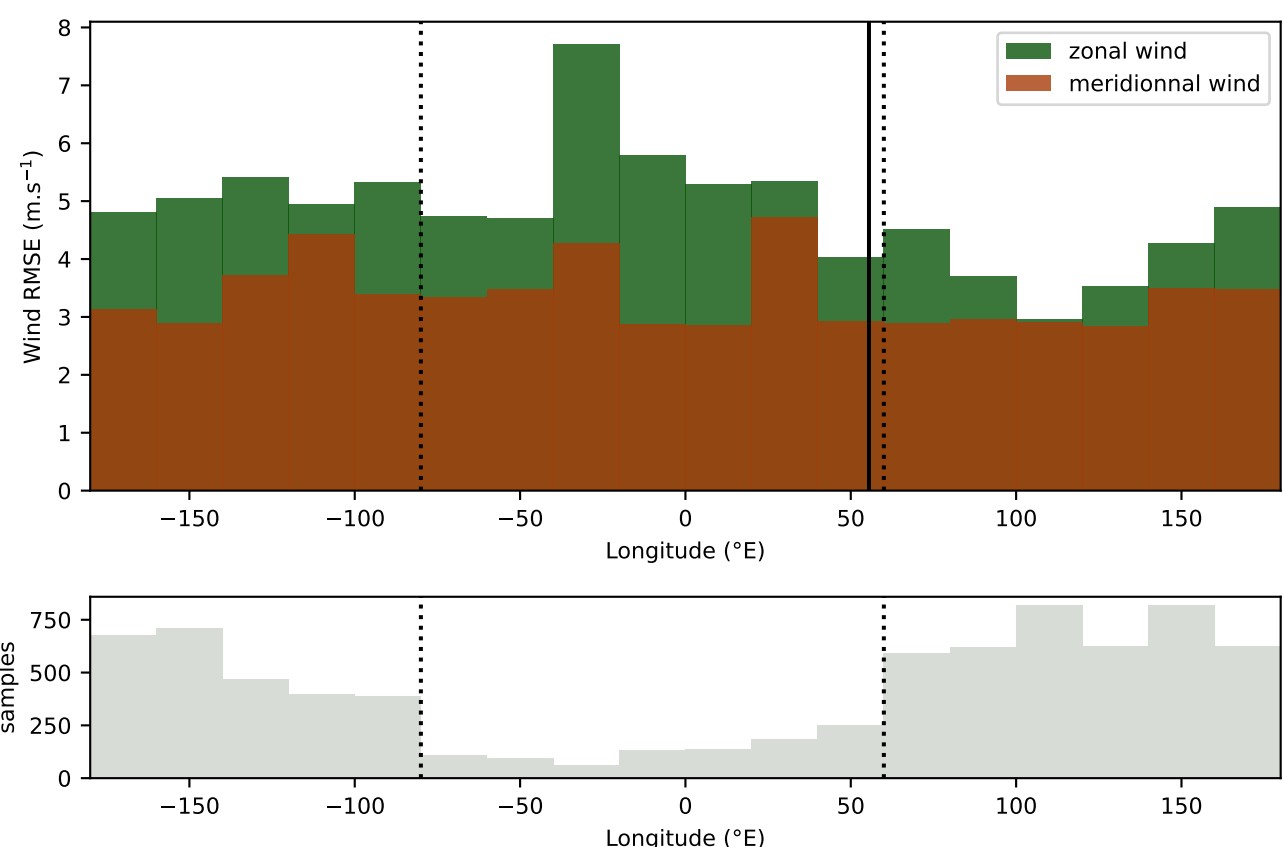

**Figure 4.** The figure at the top shows the distribution of root mean square error (RMSE) of zonal (green) and meridionnal (orange) wind from ERA-5 as a function of longitude. Only data between 8°S and 8°N are shown. The figure at the bottom gives for each longitude the number of samples available. The solid black line indicates Mahé's latitude. The dashed line highlight the longitudes between which there is a lack of observations.





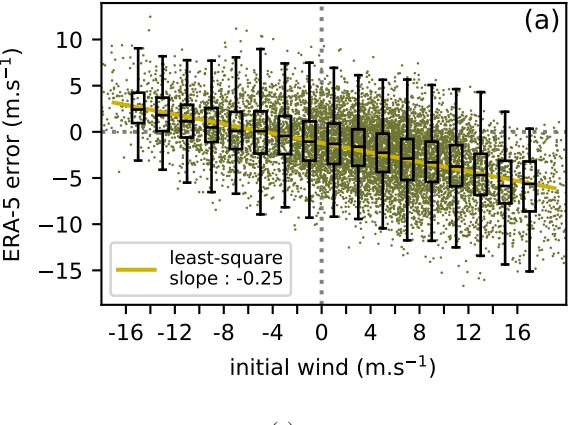
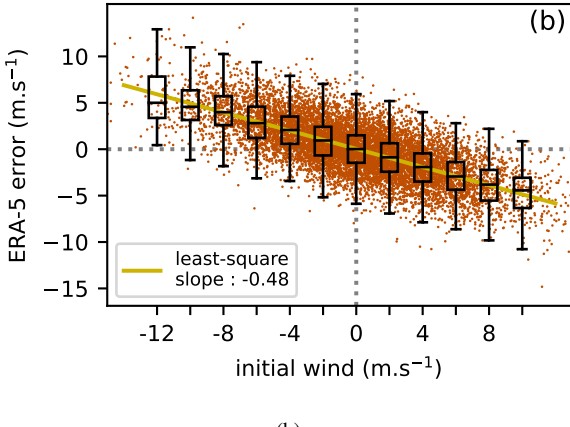

(a)  (b)

**Figure 5.** Scatter plot of the ERA-5 wind error as a function of the wind, in the zonal direction (panel (a)) and in the meridional direction (panel (b)). The black boxplot purpose is to visualize the scatter plot distribution. The grey dotted line show the null wind and null error axis.

be $86400 \times 2.09 \sim 181\,10^3$ m, or about $-1.65$ degrees after 24 hours, and about $-362$ km or $-3.3$ degrees after 48 hours. Importantly, one should keep in mind that this is only an upper bound, obtained with the exaggerated assumption that errors in the zonal wind persist coherently and do not vary for a day or more. Now, it turns out that the bias in longitude is indeed found to be comparable to this upper bound, lower but only by a third: $-2.5$ degrees after 48 hours.

Interestingly, integration over a day has accentuated the differences relative to a Gaussian distribution, and this becomes even more evident after an additional day (see panel c). This should be interpreted in light of expectations for the effects of two different processes:

1. there are errors that retain a coherency over a timescale of a few days, and may correspond to a bias in the mean winds or to missing Kelvin waves. Resulting errors grow linearly in time.

2. there are errors that vary on shorter timescales, corresponding to (the zonal component of) fast varying motions, typically different equatorial waves. Integrated over time, these errors partly cancel out. In the case of a process analogous to a random walk, one expects a growth of the error scaling with the square root of time.

The first type of errors, coherent in time over timescales of a day or more, strongly contributes to the error in longitude.

For the error in latitude, there is evident similarity between the the shape of the error in latitude after a day of trajectory
(Fig. 6b) and that of errors in the meridional wind (Fig. 3b). Again, the distribution does not fit the gaussian curve with same mean and standard deviation, but the difference is more pronounced: indeed there are significantly more outliers and the kurtosis value reaches 1.0.

The error in distance after 24 hours of trajectory is displayed in Fig. 6d. Large errors are found, implying caution in the use of modelled trajectories in the equatorial stratosphere: for only half of the trajectories, the error is less than 260 km. For a tenth
of the trajectories the error is greater than 600 km.



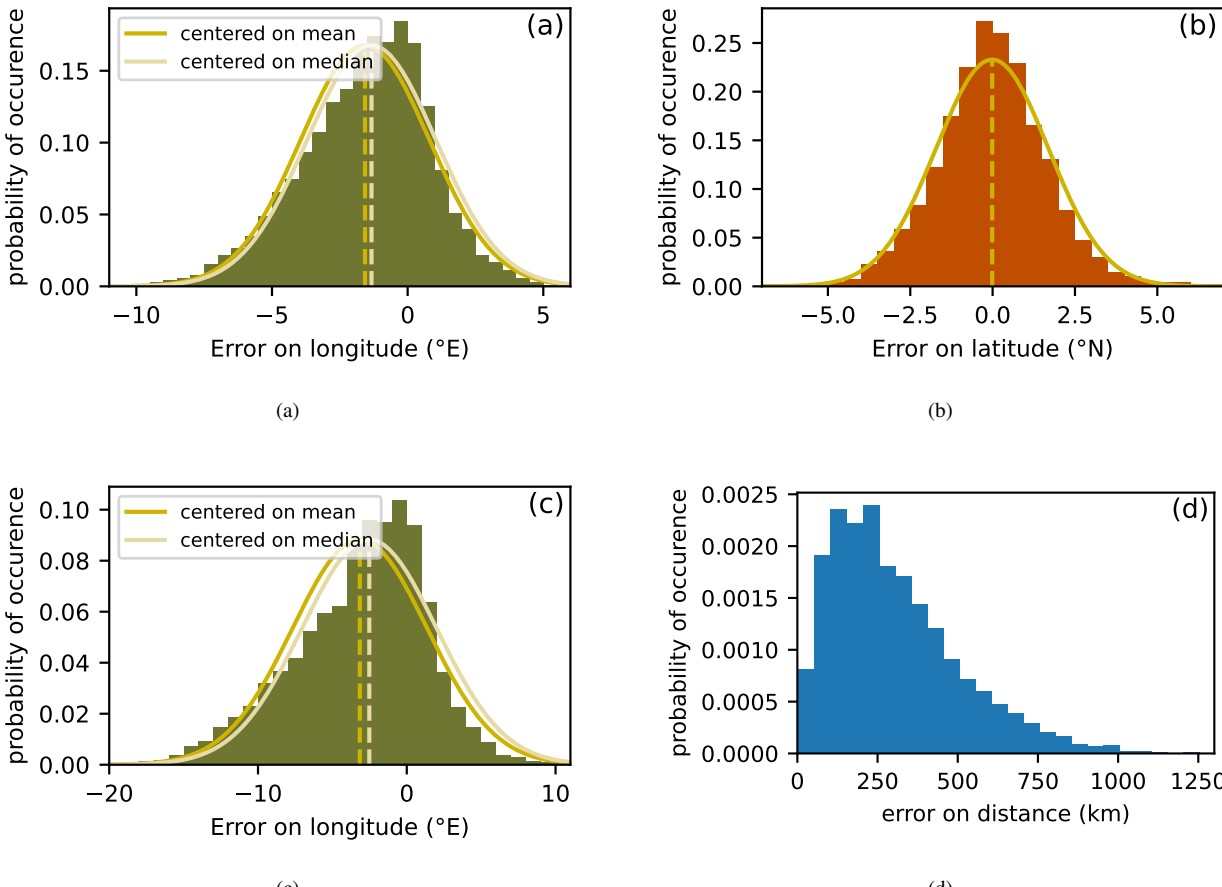

**Figure 6.** Error on 2019 and 2021 balloon positions after a day using modelled trajectories launched every three hour: panel (a) for longitude, (b) for latitude and (d) for distance. Panel (c) shows the error longitude after two days. The yellow curves are the a Gaussian distributions with the same means and standard deviations as the underlying distributions. The orange curve is a Gaussian with the same median and standard deviation. (b) shows the error in degrees of latitude after a one day forecast. Bin sizes are for panel (a) : 0.5°E, for panel (b) : 0.5°N, for panel (c) : 1°E, for panel (d) : 50 km.

| | 2019 Strateole campaign | 2021 Strateole campaign | 2019 and 2021 campaign |
|---|---|---|---|
| median | 0.007° | −0.040° | −0.027° |
| mean | 0.022° | −0.048° | −0.013° |
| standard deviation | 1.742° | 1.681° | 1.713° |
| kurtosis | 0.951° | 2.734° | 1.769° |

**Table 3.** Key statistics across different campaigns of meridional position error after a day





| | 2019 Strateole campaign | 2021 Strateole campaign | 2019 and 2021 campaign |
|---|---|---|---|
| median | $-1.361°$ | $-1.317°$ | $-1.332°$ |
| mean | $-1.623°$ | $-1.520°$ | $-1.57°$ |
| standard deviation | $2.436°$ | $2.317°$ | $2.379°$ |
| kurtosis | $0.260°$ | $0.355°$ | $0.314°$ |

**Table 4.** Key statistics across different campaigns of zonal position error after a day

The persistence in time of errors in winds is further investigated, as a potential source of information on the uncertainty of trajectory forecasts. As suggested by the trajectories of Fig. 2, the evolution of a modelled trajectory remains consistent over a day or more: in other words, many poor modelled trajectories are poor *from the start*, because the initial wind is already wrong, not because it progressively diverges after several hours or half a day. To explore and confirm this, Fig. 7a) shows how the error

in longitude after one day in modelled trajectories relates to the error on the initial zonal wind. It turns out that there is a strong correlation and that the strongest errors are negative, consistent with the negative errors in the zonal winds. Knowing the error in the zonal wind at the initial time of a trajectory, one already has a reasonable estimate of the error to expect after 24 hours in a calculated trajectory. This has implications for trajectory forecasts.

In contrast, there is a much weaker relationship between the initial error in meridional wind and the error in latitude after

a day of modelled trajectory, as shown in Fig. 7b). The points in the scatter plot remain far from the estimate obtained by assuming a constant error in the wind (orange line). The error in latitude after 24 hours appears dominated by 'random' errors occuring over the course of the day.

The difference in behaviour relative to the error in longitude is consistent with two differences between zonal and meridional winds: the former has a non-zero mean, whereas the latter essentially oscillates around a null value. The families of waves

contributing to variations in the meridional wind do not include Kelvin waves, which will have long timescales. These two differences are strikingly illustrated in the different aspects of Hovmoller plots of the zonal and meridional winds along the Equator, e.g. Fig. 2 of Podglajen et al. (2014a).

The strong errors in the winds, and hence in calculated trajectories, reflect the difficulty in modelling winds near the Equator. As the thermal wind relationship does not apply in the vicinity of the Equator, satellite observations of the temperature field

do not constrain the wind field. Figure 8 displays the distribution of errors in distance after 24 hours of modelled trajectories, as a function of the initial latitude of the trajectories. There is a clear contrast between the vicinity of the Equator, for initial latitudes in the range of 7°S to 7°N, and outside. In the equatorial belt, mean errors are typically of the order of 300 km, and errors larger than 500 km commonly occur. Outside, the sampling is much poorer but suggests errors less than 200 km, without notable outliers. Additionally, the errors associated to the two different flight levels were plotted separately, but this does not

reveal any systematic difference.



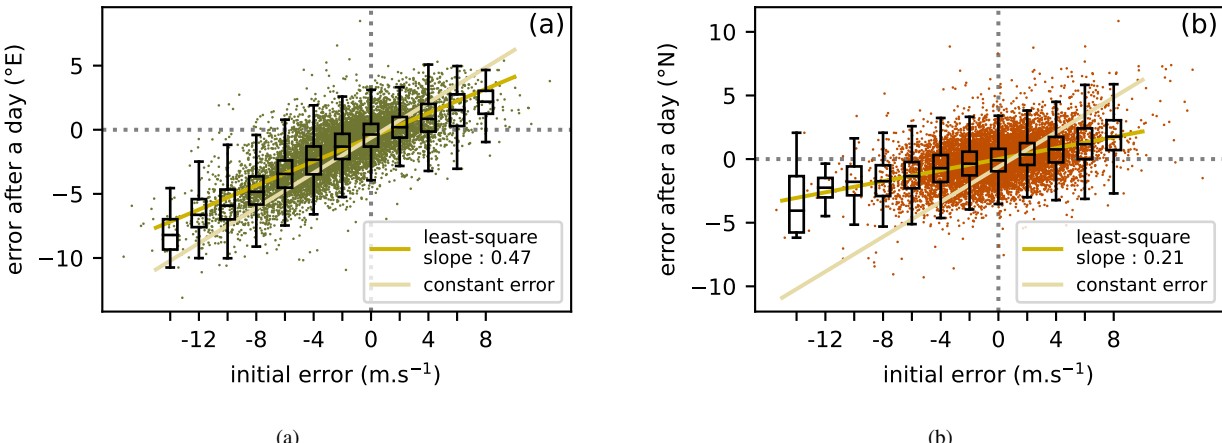

(a)                                          (b)

**Figure 7.** Scatter plot of the position error in the modelled trajectories as a function of the error on the initial ERA-5 wind. The black boxplot purpose is to visualize the scatter plot distribution. The yellow line is the error if the initial error stayed the same. The grey dotted lines show the null wind and null error axis.

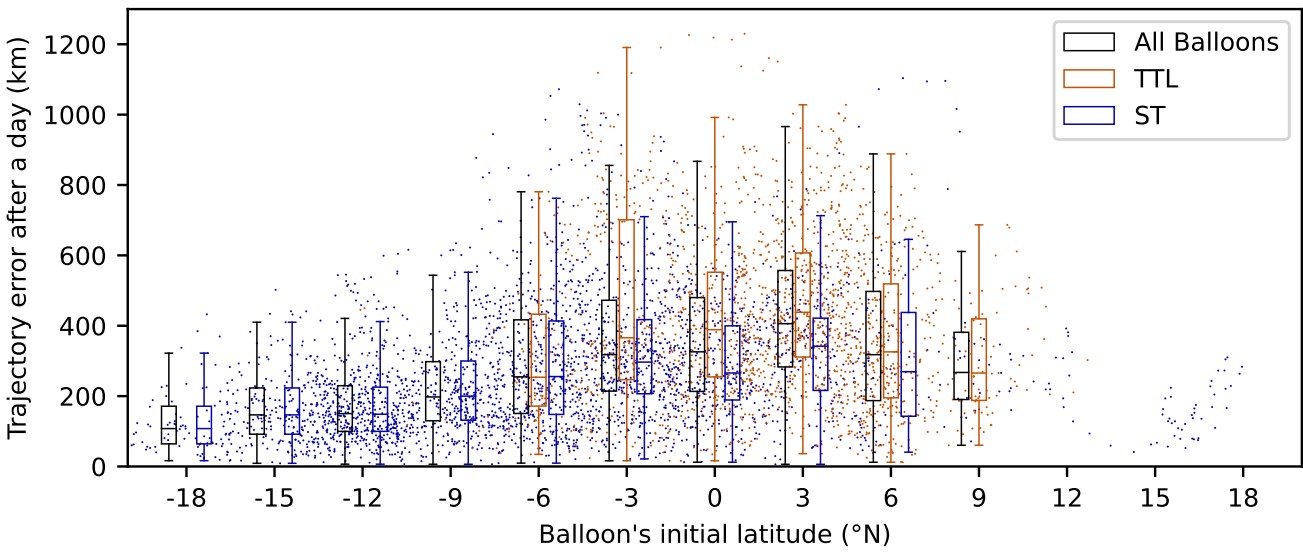

**Figure 8.** Scatter plot of the initial latitude from 2019 Strateole 2 campaign's balloons and the error in distance after 2 days. The boxplot corresponds to points in a range of 4 degrees of latitude.





| | 2019 campaign | | 2021 campaign | |
|---|---|---|---|---|
| | forecasted trajectories | modelled trajectories | forecasted trajectories | modelled trajectories |
| longitude mean | −1.582 | −1.623 | −1.562 | −1.519 |
| longitude standard deviation | 2.473 | 2.436 | 2.794 | 2.317 |
| latitude mean | −0.007 | 0.021 | 0.070 | −0.048 |
| latitude standard deviation | 1.928 | 1.742 | 1.886 | 1.681 |

**Table 5.** Mean and standard deviation of latitude and longitude error from forecasted and modelled trajectories after a day.

## 3.3 On trajectory forecasts

Operationally it is important to forecast the balloon's position, for safety and potentially for operating certain instruments. Acknowledging the fact that the skill of calculated trajectories can vary wildly (Fig. 2), a key issue is to anticipate the accuracy of a forecast trajectory.

During both campaigns, trajectory forecasts were calculated very regularly based on the deterministic forecasts of the ECMWF IFS model, which are provided twice per day. For each balloon, trajectory forecasts are initiated about every hour, starting from the most recent balloon position and using the most recent forecast available. The trajectories are integrated for five days, but we have only considered errors after 24 hours as these are already large. The errors in position after 24 hours were similar to those obtained for modelled trajectories, as shown in Table 5. The biases and the standard deviations are similar

between forecast and modelled trajectories. The standard deviations are only about 10% larger for forecast trajectories. In other words, for a lead time of 24 hours, errors in forecast trajectories are dominated by errors that are also present in the analyses, and forecast trajectories are therefore not discussed further in the present study.

## 4 Motions of air masses in the vicinity of the balloons

A second purpose of this study is to investigate the motions of air around the balloons, in order to provide insights for the

interpretation of the measurements made by the different instruments onboard Strateole 2 balloons. At the balloon level, measurements have the very attractive feature of being quasi-Lagrangian. For the analysis of atmospheric dynamics, this has the advantage of giving direct access to the intrinsic frequency of motions, facilitating the analysis of gravity waves (Vincent and Hertzog, 2014; Jewtoukoff et al., 2015; Podglajen et al., 2016a) and making it possible to distinguish the contributions of high versus low frequency waves (Corcos et al., 2021). Regarding scientific objectives other than dynamical, and more

specifically regarding aersosols, clouds and transport, the quasi-Lagrangian behavior of the balloons suggests the possibility of interpretations close to our Lagrangian understanding of processes within an air parcel (e.g. Jensen et al., 2012). In practice, there are however a number of difficulties, minor ones for measurements at flight level, and significant ones for measurements below the balloon, as discussed below.





First, for measurements made at the balloon level, two minor difficulties may need to be taken into account:

    – the balloon undergoes small oscillations around its equilibrium position. These occur at a period slightly shorter than the buoyancy frequency (typically 3 minutes). The vertical excursions cover a few tens of meters, their amplitudes vary along each flight. Use can be made of these vertical excursions to estimate the local stability and make inferences on turbulence, see Wilson et al. (2023) for a more detailed discussion.

    – on larger scales, the balloon trajectory may differ from isentropic air parcel trajectories because the balloon follows
    an ispoycnic behaviour. When a gravity wave is present for instance, the vertical displacement of isopycnic surfaces is less than the displacement of isentropic surfaces, implying that the balloon is scanning through a certain range of isentropic surfaces. As the balloon ascends, surrounding air parcels are ascending faster and the balloon is sampling air from below in fact, see discussion by Carbone et al. (2024) for a discussion and implications when interpreting water vapour measurements.

In other words, a superpressure balloon is moving with the horizontal flow (quasi-Lagrangian behaviour), but is sampling a small vertical extent of the air mass in which it is drifting: a few tens of meters due to its neutral oscillations, and about ten to twenty K in potential temperature due to the difference between isopycnic and isentropic vertical displacements. This matters for interpretation of *in situ* measurements, but should have minor impacts on trajectories (Podglajen et al., 2020). To confirm this, the differences between isopycnic and isentropic trajectories, in the horizontal, are investigated in Sect. 4.1.

    For the measurements below each balloon, other and greater difficulties are present: differential advection by sheared winds implies that air parcels at different levels disperse over time. It is challenging to distinguish between variations attributed to spatial inhomogeneities and those attributed to temporal evolution. Such attribution and interpretation requires information both about the dispersion in time of air parcel trajectories, and on the scales of the phenomenon observed. Section 4.2 provide insights on the former.

## 4.1 Isentropic and isopycnic trajectories

The balloon trajectories are isopycnic, whereas air parcel trajectories, on timescales for which trajectories may be considered adiabatic, are isentropic. To document their differences, isentropic and isopycnic trajectories have been calculated, initiated from the balloons' positions every hour over all balloons during 2019 campaign. The distance after 24 hours between the two types of trajectories is obtained and displayed in the left panel of Fig. 9, to be contrasted with the errors between modelled and
real trajectories, shown in the right panel of Fig. 9.

    After 24 hours, the distance between an isentropic and an isopycnic trajectory is at most about 150 km, and its average is only about 40 km while the median is less than 30 km. This is a very small fraction of the distance covered in 24 hours: over the two campaigns, the mean distance covered in 24 hours is 664 km, with a standard deviation of 368 km, and the median is 622 km.

    For comparison, the distances between the modelled trajectories (whether isopycnic or isentropic) and the real balloon trajectory is also reported (Fig. 9b): the mean error is near 270 km, and errors can reach more than 1000 km. The distributions





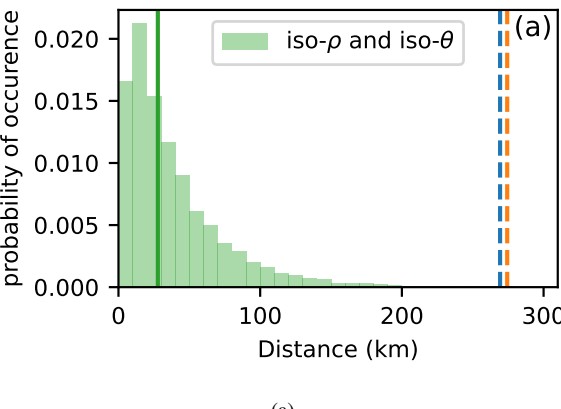
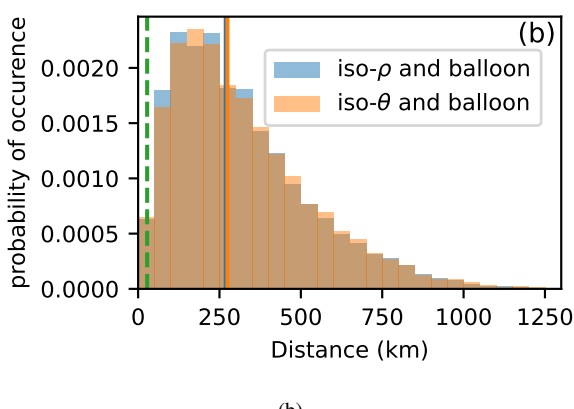

(a)                                   (b)

**Figure 9.** Probability Density Function of the distance between trajectories after one day. Panel (a) shows the distance between ispoycnic and isentropic trajectories started from a balloon's position, with the green solid line indicating the mean. Panel (b) shows the distance between modelled trajectories and real trajectories, with the solid lines indicating the means (blue for isopycnic trajectories, orange for isentropic trajectories). Dashed lines recall, in the other panel, the means of the other distributions, for comparison. These distributions were calculated for trajectories of the 2019 campaign. bin size : panel (a) : 10 km, panel (b) : 50 km.

of differences between calculated (isentropic and isopycnic) and real balloon trajectories are nearly indistinguishable, only a minor difference is found with errors of isopycnic trajectories slightly weaker than isentropic ones, as expected. In other words, the error in modelled trajectories overwhelmingly results from errors in modelled winds, not from the description of

the balloon's flight physics. In practice, one may also retain that isentropic trajectories, in terms of their projection onto the horizontal, will be very similar to ispoycnic trajectories, confirming expectations. Finally, one may note that the distribution of errors after 24 hour trajectories calculated for the 2019 alone (Fig. 9) is very similar to the distribution of errors calculated for both campaigns (Fig. 6), which is consistent with the similarity in the errors on the winds.

## 4.2 Isentropic motions below the balloons

Greater difficulties arise for measurements which observe the air below the balloon. Indeed, several instruments have been developed specifically to obtain information below the balloon (e.g. Ravetta et al., 2020; Kalnajs et al., 2021) or profiles in the vicinity of the balloon (e.g. Cao et al., 2022). For simplicity and because measurements made below the balloons can include information relevant for cirrus clouds (presence of particles, water vapour), we focus on motions of the air below the balloon. The shear between the flight level of the balloon and air below prohibit an interpretation of measurements as quasi-Lagrangian

time series. Nonetheless, the velocity difference between the balloon flight level and the altitude of potentially interesting measurements is about two orders of magnitude weaker than the typical speed of airborne measurements ($\sim 200 \, \mathrm{m\,s^{-1}}$). Interpretation of measurements below the balloon are intermediate between time series sampling the evolution of an air mass, and instantaneous profiles sampling the spatial variations. The interpretation will depend on the scales of the phenomenon




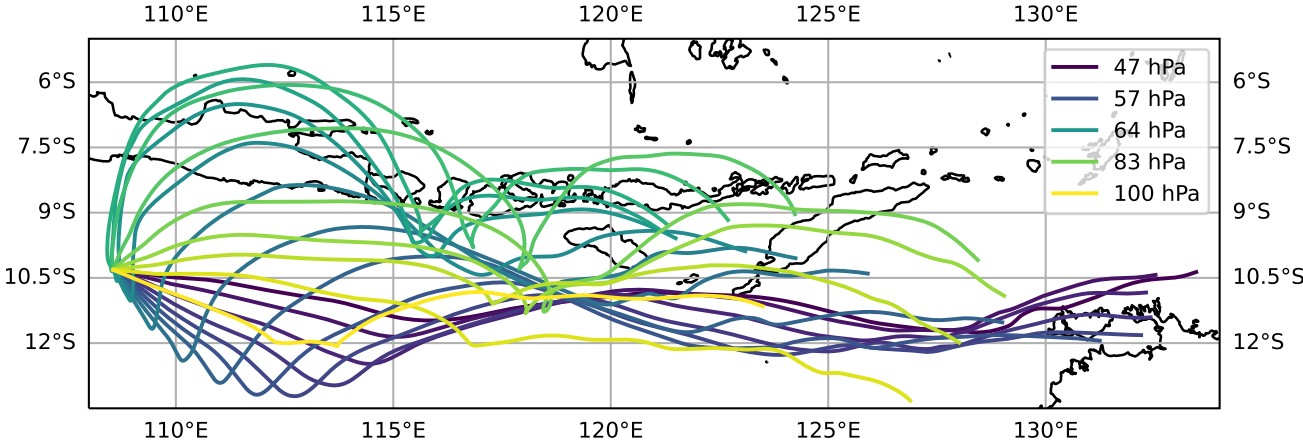

**Figure 10.** Three-day long isentropic back trajectories starting the 26[th] of January from the position of balloon 1 at several pressure level.

observed, and on the velocity difference between the level of the balloon and that of the measurement. The remoteness and
multiscale character of processes in the tropical UTLS provide motivation to investigate this issue carefully: for instance, even
modest insights about the lifecycle of cirrus clouds near the tropopause would constitute a valuable outcome (Lesigne et al.,
2023), given the difficulty of extracting information from existing observations (Noel et al., 2018).

The purpose of the present section is to investigate the origin of air parcels that are aligned below the balloon at a given time.
Vertical shear and the spatio-temporal variations of wind can lead to significant dispersion, with variations in time as illustrated
in Fig. 10. Depending on the spatial scale of the feature observed below the balloon, this may be crucial or indifferent: in situ
cirrus clouds may reach dimensions larger than 1000 km (Taylor et al., 2011; Podglajen et al., 2016b).

Isentropic back trajectories have been systematically calculated for the 2019 campaign, starting from positions vertically
aligned with the balloon positions every twelve hours. For the initial altitudes, it has been chosen not to set altitude levels
relative to the balloon position at a given time (e.g. every 250 m below the balloon level, down to 3 km). Whereas this approach
could be convenient for a specific case study investigation, it is unfavorable for general statistics about the relative dispersion of
the trajectories. Indeed, statistics about the dispersion of trajectories, say at balloon level and 500 m below, would mix different
levels in the atmosphere. Moreover, the balloon observations are not directly used here, the purpose is to document the relative
dispersion due to vertical shear, not to illustrate again the divergence of real and modelled trajectories due to model errors.

For the initial altitudes, two options were used: one consists in selecting isentropic levels and the other in selecting pressure
levels. The ranges are chosen in order to encompass the balloon flight levels and to extend below. These trajectories allow to
quantify the relative dispersion of parcels at different levels, for which several diagnostics can be used. Figure 11 illustrates
the time evolution of two key statistics of back trajectories started on different isentropic levels (from 378 K to 438 K, framing
the flight levels of the balloons, roughly from 15.5 to 20.5 km): the mean and standard deviation. The median and the 90th
percentile follow similar behaviours, albeit with different magnitudes (not shown). The separation between the back trajectories





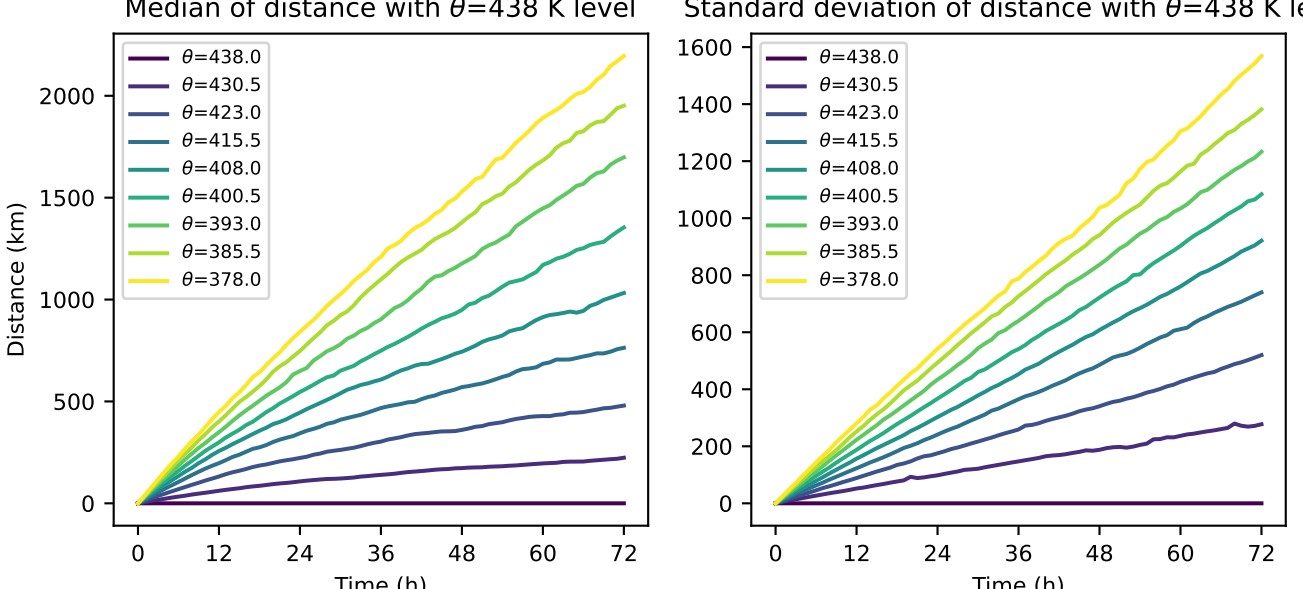

**Figure 11.** Key statistics describing the dispersion of air parcels below the balloons, for the first three days of isentropic back trajectories initiated at potential temperature levels between 378 K and 438 K. The right panel shows the median, the left one the standard deviation.

grows in time, in an approximately linear way on this timescale of three days, and grows systematically faster for larger vertical separation, as could be expected. Large values of horizontal separation are reached: for a vertical separation of 30 K, the median and mean separation after only a day are of order 500 km. The plot for the 90th percentile (not shown) indicates that for 10% of cases the horizontal dispersion exceeds 1000 km. These orders of magnitude emphasize the need to carefully assess shear and dispersion for the interpretation of measurements below the balloons. The very large values reached for the standard deviation, more than 1000 km after three days, emphasize the variability in the situations described.

In the following paragraphs, we investigate the relationship between the dispersion of back trajectories and the shear: this could help to assess, from a limited knowledge of the flow, an order of magnitude of the dispersion. The evolution of the dispersion in time is first considered, then the sensitvity to vertical separation.

A first variation to document is the variation in time: if the differences in wind between a reference and another level are simply constant in time, the distances between back trajectories would simply scale linearly with their duration, $t$. At the other extreme, if the wind differences were random and with variations short relative to the durations of the back trajectories, one would expect dispersion scaling with $\sqrt{t}$. To explore the behaviour of the back trajectories, we consider the distance $D_{\theta_r}(t, \theta)$ between a backtrajectory at level $\theta$ and that at a reference level $\theta_r$, at time $t$, see Eq. (1). This distance grows with time; at initial times, it grows linearly with time and proportionally to the velocity difference between the reference level and the level considered. We explore how this separation evolves with time, and choose to convert these growth rates (equivalent to





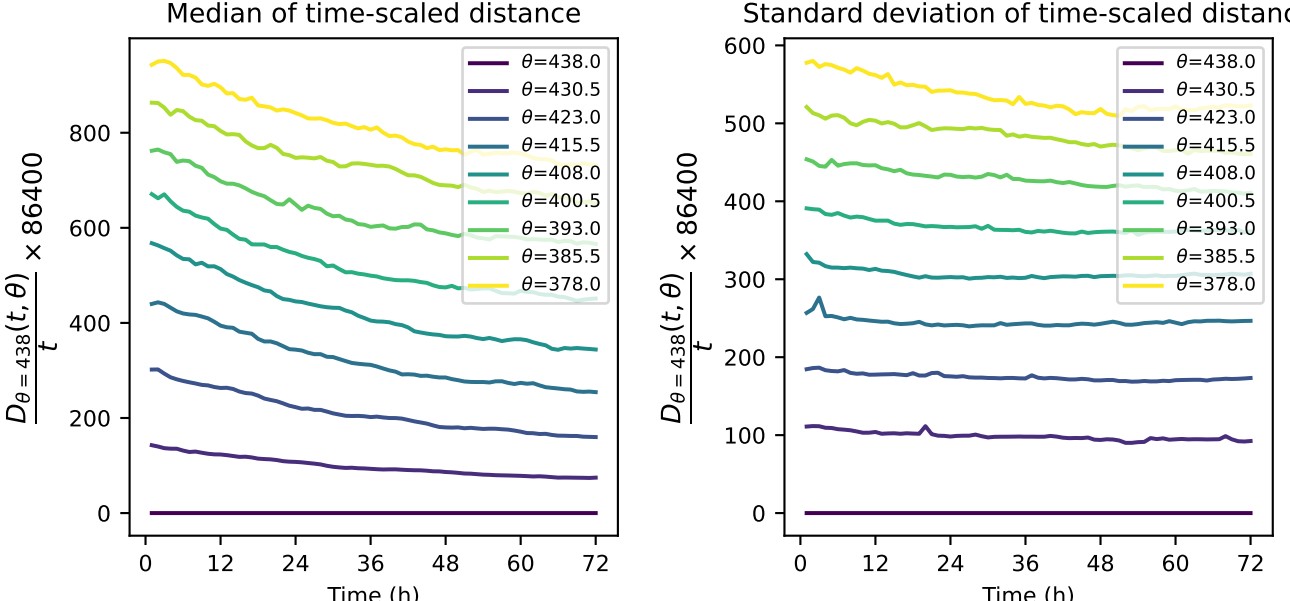

**Figure 12.** Median (left panel) and standard deviation (right) for the time-scaled distance $\tilde{d}$ (eq. (2)) of separation between isentropic back trajectories initiated on different isentropic levels below the balloons, for durations out to three days.

a velocity) to a distance, i.e. the distance covered in one day at that velocity, $\tilde{d}(t,\theta)$, see Eq. (2). In other words, $\tilde{d}$ describes the expected distance after one day for a separation occuring at the average velocity difference between the two levels. Two statistics for $\tilde{d}$ at different isentropic levels are presented in Fig. 12. The evolution in time, especially for the standard deviation, is rather flat, indicating that this linear scaling with time is a good indication of the expected separation of trajectories at different

levels. It is of course an overestimate, provides and upper bound, and the median and mean are seen to decrease with time, all the more as the vertical separation is larger.

As a complementary presentation of this rescaling, and as an opportunity to present results from the trajectories calculated on pressure levels, Fig. 13 displays the PDFs of rescaled distances, $\tilde{d}$, between air parcels at different vertical separations in pressure, after one, two and three days (see Eq. (2)). On these timescales, it appears that wind differences between different

levels have sufficient persistence in time for the dispersion to scale nearly linearly with time. As stated above, the linear scaling provides an upper bound: distances after two days are a bit shorter than twice the distances after one day. The difference is more pronounced as the vertical separation increases. However, the difference are rather slight, and a first conclusion of these two figures, 12 and 13, is that on timescales of a few days and for separations less than two kilometers, a linear scaling in time constitutes a sound, first approximation.

A second variation to document is the variation with vertical separation. Here again, a point of reference can be obtained by assuming that the difference in winds between two levels remains constant over the duration of the back trajectories. However,



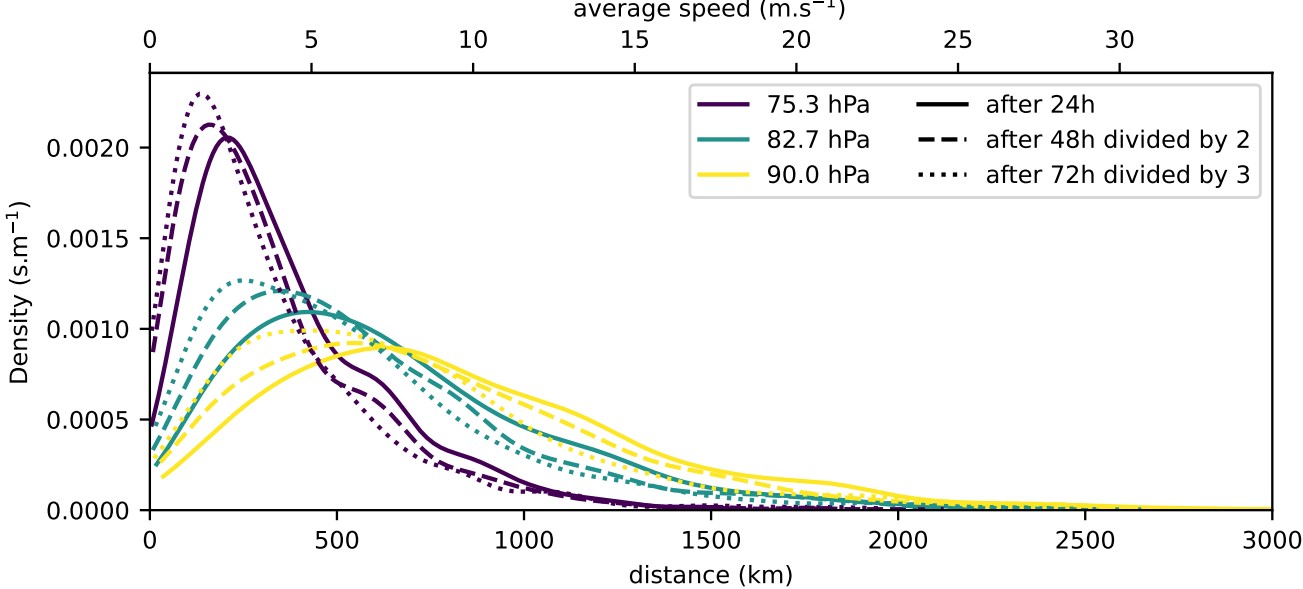

**Figure 13.** Probability density of the separation between an air parcel at 68 hPa and air parcels at three levels below (purple lines for 75.3 hPa, green lines for 82.7 hPa, yellow lines for 90 hPa). The solid line is for the distance after 24 hours. The other two PDFs have been rescaled and hence correspond to PDFs of $\tilde{d}$: the dashed lines is for the distance after 48 hours divided by 2, the dotted lines is for the distance after 72 hours divided by 3.

a complication comes from the fact that the vertical shear varies in time and location. On a timescale of 24 hours, it may be expected that the differences in velocity between two different vertical levels change only moderately. In that case, the separation after 24 hours should scale with the shear between the levels considered. The separation distance between two back
trajectories is thus rescaled with the initial shear between the two levels and for that starting date, as $\hat{D}$, see Eq. (3). This has been carried out and key rescaled statistics are presented in Fig. 14. It is interesting to first examine the time evolution of the rescaled averages and rescaled standard deviations. After an initial 12 to 24 hours which include rapid variations, the standard deviations seems to settle around values between 0.8 and 1.4, without a clear ordering relative to vertical separation. The average distance remains of order 1 or a bit less (down to 0.85 after three days), with lower average values for shorter
vertical separation. The two quantiles that are represented in complement (the median and the 90th quantile) provide elements to understand this evolution: the 90th percentile increases sharply in the first 12 to 24 hours, then stabilizes around values between 1.6 and 1.8, without any clear ordering relative to vertical separation. The median continuously decreases, down to values between 0.5 and 0.8 after three days, with a clear ordering: the shorter the vertical separation, the faster the decrease.

We propose the following interpretation:

– for a majority of trajectories, the initial shear, measured in absolute value, constitutes an overestimate of how trajectories that are vertically separated will disperse. Part of the initial shear indeed corresponds to relatively short lived perturba-





tions or waves: the time evolution of this component of shear will lead to dispersion much less than expected from the initial value of shear. This is reflected in the continuous decrease of the median. Moreover, for oscillations, there may be cancellation as we integrate in time. These short-lived components of the shear contribute more in the short vertical scale signal than in at longer scales, which presumably explains why the median decreases faster for shorter vertical separation.

– we are dealing with a strictly positive quantity: distances. The shear experienced by air parcels over their back trajectories varies relative to the initial shear. For some of these, the initial shear may have been weak, and our rescaling expects only very modest dispersion: in the first day or half day, any finite shear may lead to a dispersion significantly larger than the expected one. This explains the growth of the 90th percentile. There is in this case no particular relation to the vertical separation, hence the absence of ordering with respect to the vertical separation.

– the effect of the cases for which the initial shear leads to an underestimate of the dispersion dominate on short timescales, hence the transient (and modest) growth of the rescaled mean distance. Over longer times (more than half a day), it is the behavior of the majority of cases in which the initial shear has overestimated the dispersion which dominates: the rescaled mean distance decreases with time, all the more as the vertical separation is weak and the initial shear may have included short-scale, short-lived wind variations.

## 5 Discussion and Summary

Superpressure balloon measurements of the first two campaigns of the Strateole 2 project (Haase et al., 2018) and the ERA5 reanalyses (Hersbach et al., 2020) have been used to investigate winds and trajectories in the equatorial lower stratosphere, with two objectives. The first was to assess the accuracy of the reanalyzed winds, and the second was to document the 'dispersion' of back-trajectories, for air parcels vertically aligned below a balloon and within a couple kilometers.

The accuracy of reanalyzed winds has been assessed by interpolating ERA5 wind to the balloon locations. A previous assessment had used three balloons from the Pre-Concordiasi campaign (in boreal spring of 2010, Podglajen et al. (2014a)), with operational analyses and reanalyses available at the time (operational analyses of the ECMWF, ERAI and MERRA reanalyses). Somewhat surprisingly, the distribution of wind errors has only seen a modest improvement:

– the largest wind errors are found on the zonal wind, with a bias of $-2.09\,\mathrm{m\,s^{-1}}$ and a standard deviation of $3.76\,\mathrm{m\,s^{-1}}$. Relative to the 2010 campaign, these values constitute a decrease of 12.5 and 20%, respectively (see table 1). The bias needs to be taken with caution: it reflects missing Kelvin waves, probably more than a bias in the zonally averaged wind.

– errors on the meridional winds remain unbiased, but the standard deviations are only a bit weaker, $3.24\,\mathrm{m\,s^{-1}}$, representing only a 10% improvement relative to 2010 (see table 2).

Reanalyzed winds were also used to calculate trajectories of *virtual* balloons, or more simply isopycnic trajectories to be compared with the real balloon trajectories. These provide further insights on the wind errors, with practical implications for





**Figure 14.** Some key statistics of distance with the air parcel below the balloon scaled by the initial shear (eq. (3)): median (top left), mean (top right), standard deviation (bottom left) and 90[th] percentile (bottom right).





the uses of trajectory forecasts during future campaigns. The comparison between virtual and real trajectories revealed or confirmed the following:

1. in the tropical band and for timescales of a day or a couple of days, forecast trajectories (calculated with forecast winds) and modelled trajectories (calculated with analyzed or reanalyzed winds) have very similar errors (standard deviation only 10% larger for forecasts after 24 h). This differs relative to the mid- and high latitudes where reanalyzed winds are much more accurate (Selvaraj et al., 2019).

2. the geographical dependence of errors is mostly as expected: the strongest errors are found, fairly homogeneously, in an
equatorial band from 7°S to 7°N (cf Fig. 8).

3. errors in the zonal wind component have a consistency in time such that the longitudinal error after 24 hours strongly correlates with the initial error in the zonal wind (cf Fig. 7). In practice, this implies that the zonal wind error at the time of initiation already provides a fair indication on the uncertainty of forecast trajectories. Fundamentally, this illustrates that the zonal wind errors are tied to the background wind and/or to low-frequency waves such as Kelvin waves, which
remain coherent over timescales of a day or a few days.

4. the errors in the latitudinal position after 24 hours is very weakly correlated to the initial error on meridional wind (cf Fig. 7). This is consistent with the errors in meridional winds being tied to higher-frequency waves, with signatures that may oscillate within a day.

5. The clear correlation between meridional wind and error on meridional wind (cf 5) provides evidence that equatorial
waves are present in the reanalysis, but underestimated, in line with previous findings (Kim and Alexander, 2015).

In interpretations of balloon-borne observations of phenomena within a couple kilometers below the balloons, one commonly needs to evaluate the origin of the air mass that is sampled. Virtual air back trajectories were calculated to document the separation between a balloon and of air at different levels below the balloon. The main findings are

– over a period of up to three days, the growth in time of the distance between trajectories at a reference level ($\theta$ =438 K)
and levels below is reasonably approximated as linear;

– situations however vary greatly in time, as the vertical shears vary. This implies that the vertical shear present between the balloon level and an observed phenomenon below the balloon needs to be carefully assessed in order to interpret variations in the phenomenon. Cases of weak vertical shear will favor the observation of temporal evolution (e.g. of a wide cirrus cloud). Cases with significant shear will require interpretations in terms of temporal *and* spatial variations.

An important limitation comes from the use of reanalyzed winds to calculate all trajectories. Part of the errors in reanalyzed winds come from underestimated wave variability, especially at relatively short scales. To leading order, wave fluctuations should average out in trajectory calculations. Confirming this would require further study, and high-resolution simulations including a richer field of equatorial waves would be appropriate for such investigation.



## 6   Conclusions

The above findings emphasize the need for and value of observations of winds in the lower equatorial stratosphere. Significant errors persist in reanalyses and have only weakly decreased over the past decade. The bias found in zonal wind highlights an underestimation of equatorial Kelvin waves, consistent with previous studies Kim and Alexander (2015). An implication of these persistent errors in modelled winds is for operational planning of future superpressure balloon campaigns: errors in forecast balloon trajectories of several hundreds of kilometers after 24 hours make it difficult to coordinate superpressure

balloon observations with ground-based measurements. Another implication is for process studies using reanalyzed winds to calculate Lagrangian trajectories: in addition to the challenges of representing temperature fluctuations (e.g. Jensen et al., 2016), the horizontal advection by reanalyzed winds in such studies imply large uncertainties.

*Data availability.* Strateole 2 data is available from strateole2.aeris-data.fr . The ERA5 reanalyses are available from cds.climate.copernicus.eu
.

## 510   Appendix A: Zonal wind bias induced by the motion of a balloon Kelvin waves

As explained in the main body, the sampling by quasi-Lagrangian balloons may induce a positive wind bias compared to a sampling at fixed positions in the presence of Kelvin waves. This is due to the balloon drifting in the same direction as the phase propagation of the wave when the wind is positive (Eastward). This effect, which was referred to as wave *surfing* by Podglajen et al. (2014b), only exists for isopycnic balloon trajectories, not for isentropic air parcel trajectories, and can be

quantified in terms of Stokes drift (see Podglajen et al., 2014b).

   In order to quantify the impact of balloon-like sampling on wind and temperature statistics, compared with Eulerian sampling, we computed a set of virtual trajectories in the ERA5 launched every 2 degrees in the ERA5 and integrated for 14 days. The distribution of zonal wind $U$, meridional wind $V$ and temperature $T$ sampled along the isopycnic trajectories has been compared with fixed latitude-longitude locations at the start of the trajectories (not shown). The isopycnic-trajectory sampling

does not induce any significant bias in temperature or meridional wind (according to a Student-t test). However, the zonal wind along the virtual trajectories is positively biased by about $1 \text{ m s}^{-1}$ compared with Eulerian sampling.From this exercise, we conclude that underestimated Kelvin wave activity in a model or reanalysis, which does not create a zonal mean zonal wind bias, may still result in significant negative model bias when the atmosphere is sampled along isopycnic balloon trajectories.

*Author contributions.* PC carried out nearly all the analyses, AMF carried out preliminary analyses. The analyes and results were discussed

by PC, RP, AH and AP. Trajectory calculations were carried out by PC with support from AP and AH. Writing was carried out mainly by PC and RP, with contributions from AP and AH. Overview and rereading were carried out by RP, PC, AH and AP.





*Competing interests.*  At least one of the (co-)authors is a member of the editorial board of Atmospheric Chemistry and Physics.

*Acknowledgements.*  This work was supported by the ANR project BOOST3R (ANR-17-CE01-0016-01) and by the French-American project Strateole 2 (CNES). PC is thankful for the support from MétéoFrance in carrying out this work. Data for the Strateole 2 campaigns are available from https:strateole2.aeris-data.fr




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
