# Peer review of "Strateole 2 balloons reveal persistent errors in reanalyzed winds and trajectory calculations in the tropical lower stratosphere"

_EGUsphere, 2025_

## Author Comment (AC1)

**Revision of**

**'Strateole 2 balloons reveal persistent errors in reanalyzed winds and trajectory calculations in the tropical lower stratosphere'**

P. Cadiou, R. Plougonven, A. Podglajen, A. Hertzog and A. MacFarlane November 10, 2025

We thank the reviewers for their comments and suggestions concerning the manuscript, and their careful reading of the text. These comments and suggestions have been helpful to improve the manuscript. Below we provide answers to the reviewer's comments and suggestions, and where the text has been modified accordingly.

We reproduce the reviewer's comments in black, and include our replies in blue.

**1 Reply to Reviewer #1**

In this paper, superpressure balloon measurements from the Strateole 2 campaign have been used to investigate errors in ERA5 reanalysis winds and the dispersion of trajectories in the equatorial lower stratosphere. The analysis appears sound, and the results are useful, especially in relation to forecasting trajectories for future campaigns. The paper can be considered for publication after the authors address the following comments and suggestions:

**Comments and Suggestions**

• The location of the balloon launch should be marked in Figure 1. Additionally, clarify what the different colors represent in the figure and update the figure caption accordingly.

The figure has been modified to include a cross indicating Seychelles Islands, from whichthe balloons are launched. Each color represents a balloon flight. As the balloons are considered collectively in the rest of the study, we have not added a legend for the color of each balloon. In practice, it is difficult to identify all colors unambiguously and it is not needed int his context. However, we have included explanations regarding the visual impression that there are more observations in 2019 (upper panel) than in 2021 (lower panel), although there are

more balloons in 2021. The caption now reads:

Trajectories of Strateole 2 balloon flights during (top) the 2019 and (bottom) the 2021 campaigns. Eight balloons flew in 2019 with an average flight duration of 84 days. In 2021, 17 balloons were launched but the average flight duration was only 40 days because of issues with the balloon enveloppes encountered in 2021. The balloons are launched from Seychelles Islands (55.5-4.7°E, -4.7°S) which is indicated by a cross.

- Figure 2 is interesting. What is the reason for the large differences in forecasted trajectories between the two launches that occurred during the same season?
  - The figure is illustrative of the very variable skill that can be found in the forecasts of trajectories. We have not tried to investigate precisely possible reasons for the poor modelled trajectory presented in the lower panel. One can not nonetheless that the upper panel (good agreement between the modelled and real trajectories) corresponds to a westward portion of a trajectory, between 10 and 15°S. The lower panel corresponds to a portion of trajectory closer to the Equator (6 to 14°S), at a time of weaker winds, hence when signature of waves is likely to dominate over a more zonal flow.
- Line 210: For zonal wind (panel a, not b), remove the extra parenthesis. Corrected, thank you.
- Line 180: Please discuss the importance of the Gaussian distribution of errors.
  - It has appeared to us of interest to determine if the distribution is approximately Gaussian or not. In the absence of other information, it is generally expected that this would be the case, with the advantage that two numbers indeed summarize the distribution. It turns out that the errors follow a distribution close to but not Gaussian. This can matter for the tails of the distribution. We have considered this was worth investigating and pointing out. On the other hand, these are only distribution of wind errors, extreme values do not matter because of impacts like extreme precipitation or winds. Hence we do not emphasize this point or investigate this further. We hope we have found the right balance in the text, which has not been modified.
- The authors attribute biases in wind to missing components of Kelvin wave variability in the reanalysis. What were the phases and amplitudes of the Kelvin waves during the campaign period? Is there any supporting evidence from other observations?
  - We have not attempted to quantify the Kelvin waves in the reanalysis.

The reasoning that leads us to attribute an important part of the errors and in particular the bias in zonal wind to missing Kelvin waves is based three elements:

- on the previous study, by Podglajen et al (2014). That study similarly investigated errors in winds in analyses and reanalyses. More precisly, we had investigated operational analyses of the ECMWF and the MERRA reanalysis. During the balloon flights, there were two periods during which the balloon-measured zonal winds were stronger than the modelled winds for several days. In one period, both modelled winds were similarly wrong. In the other, one model was wrong while the other showed reasonable agreement. The comparison of the two modelled fields revealed the spatial structure of the difference, attributed to a Kelvin wave, present in the ECMWF analyses but absent in MERRA 2.
- the errors in the winds are very different between zonal and meridional winds, with the errors in zonal winds being larger and displaying a negative bias, both in Podglajen et al (2014) and in the current study.
- in the appendix, we investigate how the advection of the balloons by waves impacts their sampling of winds. A priori, one expects an unbiased sampling of the winds. However, while this holds for infinitesimal waves, this no longer is valid when velocity perturbations associated to waves are finite. Advection in the direction of the propagation of the phase can lead to longer residence time, for the balloons, in a certain phase of the wave (the one corresponding to winds closer to the phase velocity). This can leads to an uneven sampling of the waves, as the balloons reside longer a certain phase of a wave. Now, as the equatorial waveguide imposes zonal propagation, this affects only the zonal wind component. Kelvin waves have significant signatures on zonal winds, maximal at the Equator, have finite or even large amplitudes. They are good candidates for such nonlinear effect on the sampling of the winds by the balloons, and their phase velocities are always eastward. One may expect that this leads to a systematic bias in the winds as sampled by the balloons, relative to winds sampled in fixed locations. The simulated trajectories confirm this, as described in the appendix.
- Rossby waves have comparable amplitudes to Kelvin waves over the analyzed latitudes. Why is the effect of Rossby waves on wind bias minimal compared to Kelvin waves?
  - From Hovmoller diagrams of the reanalysed winds during the different campaigns, very clear signatures of Kelvin waves are seen, with their

clear eastward propagation and significant amplitudes on winds. In the Hovmoller diagram for meridional winds, signatures of mixed Rossby-gravity waves are seen as wave packets with significant amplitudes. Such Hovmoller diagrams are shown in Figure 2 of Podglajen et al (2014), reproduced as Figure 1. As explained above, the Kelvin waves will have an important impact on the sampling of the winds. These different lines of evidence have convinced us that Kelvin waves will be the main type of waves to consider for the biases diagnosed from the comparison of reanalyzed and balloon-observed winds. However, we can not exlude a significant contribution of Rossby waves, in particular in other wind regimes (both scientific campaigns of Strateole 2 and also the Pre-Concordiasi campaign occured in an ending westerly phase of the QBO, with the balloon motions predominantly eastward).

- It would also be useful to provide a figure showing the errors in the direction of forecasted/modelled trajectories, similar to Figure 8. This is an interesting suggestion. The largest errors overall are obtained on the zonal wind, as shown by the distribution of errors in figures 3, 4 and 6. Geographically, the largest errors are concentrated in the equatorial band, between roughly 6°S to 6°N. A plot of the direction of errors as a function of latitude would likely show that the average of errors in this band show a negative, zonal error. We have not attempted to explore this however: a practical reason is that the first author has moved from research to an operational position in meteorology, and hence starting new plots or analyses would take some time. Another reason is that we anticipate that this would likely confirm expectations from the other analyses that have been carried out, or would require further investigations which would be too detailed for this publication.
- Line 275: The sentence "in other words, many poorly modelled trajectories are poor from the start because the initial wind is already wrong" reads awkwardly. Please consider rephrasing for clarity.

  We have rephrased and hope that this makes the point clearer:

In other words, poor trajectory forecasts are generally poor from the start. The initial wind already has significant error. The poor forecast does not result from error in the wind that appears and grows after several hours or half a day.

**2 Reply to Reviewer #2**

This manuscript provides a detailed and quantitative assessment of wind and trajectory errors in reanalyses using the highly valuable Strateole-2 balloon

Figure 1: Hovmoller diagrams of zonal wind (left panel) and meridional wind (right panel) during the Pre-Concordiasi campaign of 2010; this is Figure 2 of *Podglajen et al (2014)*, highlighting Kelvin waves (left panel) and mixed Rossby gravity waves (right panel).

data covering the equatorial and subtropical regions. Both the methodology and explanations are reasonable and presented in sufficient detail. I have no major comments, and would only like to point out minor typos and the ordering of figures and tables.

- Throughout the manuscript, both Strateole 2 and Strateole-2 are used. It would be better to unify them into a single form.

  Thank you for pointing this out, we have written 'Strateole 2' every time.
- Although this can be resolved during the proof stage, there are instances where acronyms introduced earlier are written out in full later in the manuscript (e.g., UTLS). Please check these occurrences for consistency.
  - Thank you for pointing this out, we have not been careful in defining all acronyms indeed, especially for acronyms thought to be familiar to readers, or less important. We have gone through the text and defined the following: GPS, TSEN, ECMWF, IFS, MERRRA. In doing so, we also noticed that the ERA5 reanalysis was referred to as ERA-5 in several instances, and have corrected this.
- Page 2, Line 53: "We choose to focus on winds because the errors and biases are more important than for temperature". Although a reference is provided, this statement may still be debatable depending on perspective. From the reader's viewpoint, it remains unclear why wind errors and biases are considered more important than those of temperature. A brief explanation would be helpful.

Measurements of temperature from satellites are available in the tropics as in other parts of the globe. Through hydrostatic balance, they constrain the pressure field globally. Now, in the midlatitudes, geostophic balance ensures that this information on the pressure field translates into information on the wind field. Near the Equator, as geostrophic balance is no longer valid, there is comparatively a significant lack of observations constraining wind. A previous investigation of errors on winds in analyses using three super-pressure balloons in 2010 had highlighted significant errors in winds (one event displayed a difference of more than 8 m/s during 30 consecutive days of a balloon trajectory). Measurements of temperature were also used, but revealed only reasonable errors. A sentence has been added to explain briefly the statement: Indeed, in contrast to mid-latitudes where hydrostatic and geostrophic balances constrain winds from temperature measurements, equatorial winds remain poorly constrained and prone to significant errors (Baker et al, 2014).

• Page 3, Lines 84–86: "Balloons flown at ... 'STR' (stratosphere) flights." This content is already mentioned in Lines 47–48, so you may consider deleting this sentence.

Indeed, this was redundant. We hav removed the sentence.

• Figure 1: Visually, the balloon trajectories from the 2021 campaign (bottom panels) appear fewer than those from the 2019 campaign (top panels). Please double-check that there is no issue.

Indeed, the top panel rightly gives the impression of having more balloon trajectories than the lower panel. In the 2019-2020 campaign, there were only 8 balloons but essentially all flights were successful, with several flights lasting nearly three months - the upper limit for which the balloons are qualified. During the 2021-2022 campaign, technical difficulties occured with the balloon envelopes, ending several flights prematurely. For safety reasons, most flights ended up much shorter than expected. Although 17 launches were made in the 2021-2022 campaign, the total number of flight days is a little shorter. The trajectories are also much more compact, grouped mainly over the Indian Ocean, East of the Seychelles Isalnds from which the balloons were launched.

- Page 7, Line 169: "-2.08" should be "-2.09" based on Table 1. This has been corrected.
- Page 7, Line 172: "Concardiasi" what does this mean? I could not find this word in Google.

This was a typo, this has been corrected to Concordiasi. But we have realized that the Concordiasi and Pre-Concordiasi campaigns were not properly referred to in the introduction. These are superpressure balloon campaigns in 2010, that preceded Strateole 2. The last sentence of the third paragraph of the introduction has been modified to:

These stratospheric balloon campaigns extend previous investigations at high Southern latitudes: the VORCORE campaign in 2005 (Hertzog et al, 2007) and the Concordiasi campaign in 2010 (Rabier et al, 2010). The latter campaign was preceded by the Pre-Concordiasi technological campaign in 2010, comprising three flights in the equatorial band.

• I recommend swapping the order of Figures 4 and 5, since Figure 5 is referenced earlier in the text.

We agree and have changed the order of the figures.

• Page 8, Line 210: "panel b" should be "panel a"? Indeed, we have changed to "panel a".

• Page 10, Lines 224–226: This paragraph seems to duplicate content mentioned earlier.

The fact that the sampling was sparse in 2010, and included episodes of strong errors is mentioned earlier. We nonetheless think it is useful to highlight that this will contribute to the longitudinal distribution of errors. We have therefore kept this point but shortened the sentence to make it as concise as possible. The sentence now reads:

"within the very limited sampling in 2010, notable episodes of strong errors occurred over the Indian and Eastern Pacific Oceans (Podglajen et al, 2014)."

• Tables 3 and 4 are not cited in the text. I also recommend swapping their order, since zonal wind is consistently discussed earlier in the manuscript.

The tables have been swapped, and reference to the tables has been added at the beginning of the second paragraph of section 3.2. These tables are complementary to Figure 6.

- Page 16, Line 315: "aersosols" should be "aerosols." Corrected
- Page 17, Line 325 and Page 18, Line 356: "ispoycnic" should be "isopycnic."

Thank you, this has been corrected; and thank you for indicating precisely the typo, that allowed to locate it quickly and to locate another occurrence of the same typo.

---

## Author Response (AR2)

Corrections for

**'Strateole 2 balloons reveal persistent errors in reanalyzed winds and trajectory calculations in the tropical lower stratosphere'**

P. Cadiou, R. Plougonven, A. Podglajen, A. Hertzog and A. MacFarlane

December 3, 2025

We thank the editor for accepting our study for publication and for the careful rereading. We have taken into account all the comments and suggestions, have changed text where necessary to clarify, and have completed the bibliography to include all authors for the publications with numerous authors.

Below we recall the comments and suggestions that had been made. All have been taken into account.

- Units should be written without a dot in between, thus m s-1 instead of m.s-1. This should be corrected on e.g. P8. Please check the entire manuscript.

- "table" should be written as "Table" (with a capital "T")

- Figure 10 caption: Add "on" so that it reads "starting on the 26th......"

- Figure 11: On the right figure panel the figure title is not complete (lev instead of level). Reducing the figure size a bit may fix this.

- L445: "than in at longer scales" not correct. Please check sentence and rephrase.
  We have worked on the text of the discussion to make this sentence clearer. This also concerns two points below, about lines 450-451 and 452-453. The corresponding paragraphs in the new version of the manuscript are at lines 445-457.

- L450-451: Sentence not clear, please rephrase.
  See above comment

- L452: underestimate → underestimation

- L452-453: Also this sentence is not clear, please rephrase.
  See above comment

- Figure 14 caption: "eq." Should be "Eq."

- Please check also the references. There are several errors, e.g.

- The Carbone et al. reference should be updated. The paper has been published in ACP: https://doi.org/10.5194/acp-25-10603-2025

- L565: Rather "et al." than "eds.", however for ACP all authors need to be listed.

- Podglajen et al. 2014: The a and b reference are the same. Please remove one and adjust text.

- Rabier and coauthors → also here a full list of authors should be provided

- Scaife et al: → also here all authors should be added.

- In Schoeberl et al. (2016) the doi is incomplete.